# The Mexican National Health and Nutrition Survey as a Basis for Public Policy Planning: Overweight and Obesity

**DOI:** 10.3390/nu11081727

**Published:** 2019-07-26

**Authors:** Teresa Shamah-Levy, Martín Romero-Martínez, Lucia Cuevas-Nasu, Ignacio Méndez Gómez-Humaran, Marco Antonio Avila-Arcos, Juan A. Rivera-Dommarco

**Affiliations:** 1Evaluation and Surveys Research Center, National Public Health Institute, Cuernavaca 62100, Mexico; 2Center for Research in Mathematics, Aguascalientes Unit, Aguascalientes 20259, Mexico; 3General Director of the National Public Health Institute, Cuernavaca 62100, Mexico

**Keywords:** National Health and Nutrition Surveys, public policies, overweight, obesity

## Abstract

Mexico has one of the highest overweight and obesity rates in the world. Our objective is to describe the tendency of overweight and obesity by sex, health service affiliation, and socioeconomic tertile (T1,2,3), and to give examples of public policies derived from the results of the Mexican National Health and Nutrition Surveys (ENSANUT). Data come from the 2006, 2012, and 2016 ENSANUTs, which are probabilistic surveys that allow us to make inferences at the national level, on urban and rural strata and regions; their coverage includes all the population age groups. We assessed overweight and obesity (OW + O) in all population groups. The prevalence of OW + O in preschool children was lower in T1 in all the surveys, and shows an increase by year of survey, according to the health service affiliation. In school-age women, prevalence increased over the 10 years of evaluation, in spite of the high prevalence in both genders in T3. Adolescent behavior is similar and, in adults, the prevalence of OW + O shows an increase by year of survey, gender, and affiliation, with the differences not explained by socioeconomic tertile. In conclusion, the ENSANUT series represents a surveillance system that allow us to observe the changes in overweight and obesity prevalence over the time, showing a high prevalence of OW + O in the population, and has contributed to public policy enhancement.

## 1. Introduction

The National Health Survey System in Mexico includes a series of multi-thematic surveys on health and nutrition; the health surveys are probabilistic and representative at the national level and include diverse geographic areas and population sub-groups. These surveys have given the country useful information with which to plan and evaluate the performance of the Mexican health system, since they provide precise, detailed, and representative information on the population’s state of health. Moreover, they have allowed us to estimate the coverage of programs and services, in order to quantify progress and identify new challenges.

The 1986 ENSA (National Health Survey) [1] was the first national health survey and its objective was to describe the health characteristics of Mexicans; ENSA was repeated in 1994 [2] and in 2000 [3]. In addition, in 1988 [4] and 1999 [5] two National Nutrition Surveys (ENN) were carried out by the National Institute of Public Health and financed by the Ministry of Health.

As of that date (2006), the general health and nutrition surveys were joined to form the National Health and Nutrition Survey (ENSANUT), which has been conducted three times: in 2006 [6], 2012 [7], and 2016 Mid-Stage [8]; currently, the 2018‒2019 ENSANUT is being carried out. 

Surveys in Mexico began computing the overweight and obesity (OW + O) prevalence in adult population in the 1980s [2,3,4,5]; however, from 2006 these studies were also conducted in other age groups. A high prevalence of overweight and obesity was noticed in children between five and 11 years old (34.8%), as well as in teenagers (30.6%), with roughly one in three individuals displaying these conditions. 

According to the Organization for Economic Cooperation and Development (OECD), Mexico had the highest prevalence of obesity (32.4%) in the population aged 15–74 years in 2017 [9]. Previously (2013), it had been documented that Mexico was the OECD country with the second-highest percentage of obesity: 30% of its total population. The average percentage in OECD members in this category is 22.2%. In the child population, Mexico occupies fourth place with respect to OW + O [10]; these estimations are done by using ENSANUT series data. 

In this way, ENSANUT design has favored the ability to compare estimates, which has been fundamental to gathering timely information on the magnitude of changes in the population’s health and disease patterns, as well as in nutrition. Thus, the objective of this work is to show the magnitude of the prevalence of overweight and obesity in Mexican population give examples of some relevant public health topics where ENSANUT information has been used as the basis for public policy.

## 2. Materials and Methods 

The National Health and Nutrition Survey is a national probabilistic survey with state, regional, and national representativeness for urban and rural strata (in 2016 this representativeness was only at the regional level); its design allows for comparisons between surveys.

The topics that may be studied, based on ENSANUT information, are the following: the nutritional state of children and adults in Mexico; the health status of the Mexican population and the prevalence of some chronic and infectious diseases; the population’s perception of the quality of the health system and its response; the sociodemographic characteristics of households that have catastrophic expenditures as a result of health problems; and the link between welfare programs and health and nutrition.

The ENSANUT has maintained a lifeline scheme that has allowed for the comparison of results, and included all population age groups. Therefore, all variables and measurements are obtained for the same individual, in order to determine their health and nutrition status.

The ENSANUT was a milestone since it combined two national surveys: the health survey and the nutrition survey. Its sample design is thus worth noting. The sample size of ENSANUT is based on the 2006 ENSANUT sample, which considers that the lowest proportion of importance (minimum prevalence of interest) to be estimated with precision in adults should be 8.1%. Furthermore, considering that the state-level estimators obtained by the survey should have a maximum relative error of 25%, a confidence level of 95%, a non-response rate of 20%, and a design effect of 1.7 (estimated from the 1999 Nutrition Survey and the 2000 National Health Survey), it was determined that the sample size would need to include at least 1476 households. Subsequently, in 2012, the sample size was increased to 1719 in order to over-represent households with lower economic capacity; the total number of households for the whole country thus reached 55,008. More details about the survey were published previously [6,7,8].

### 2.1. Variables and Indicators 

The topics addressed in the ENSANUT surveys consider both the follow-up aspects—in order to set trends in health and nutrition conditions—and indicators of use; they also place a particular emphasis on monitoring of prevention programs.

This paper reports published information about overweight and body mass index (BMI) in children 0–<5 years of age, school-age children (5 to <12 years), adolescents (12 to <20 years) [11] and in adults. Overweight and obesity in adults were defined according to the WHO criteria for BMI: 25.0–29.9 kg/m^2^ for overweight and 30 kg/m^2^ or greater for obesity [12].

The health care system affiliation was categorized as: none, four public systems, private, and ‘other’. Socioeconomic level tertiles were developed, derived from principal component analysis of household construction materials, urban services (water, sanitation, electricity), and household appliances [7,8].

Household food insecurity (HFI) was measured in accordance with the well-validated Latin American and Caribbean Food Security Scale (ELCSA) [13]. This scale was validated previously. The scale includes 15 questions that assess poverty-related food insecurity household experiences, ranging from being worried about not having access to enough food to going without food for a whole day, during the three months preceding the survey [14].

### 2.2. Analysis Plan

The proposed analysis focuses on obtaining the indicators mentioned above, which are constructed through average estimators and proportions of the diverse characteristics pertaining to overweight and obesity, by survey year, for the general population. In addition, we considered an analysis by different population groups (by health service affiliation and socioeconomic level). Datasets for all surveys included expansion factors (weights), which allowed the estimates to be generalized for the Mexican population. Additionally, the data were stratified according to sex and socioeconomic level. Socioeconomic level was estimated in the surveys using an imputation strategy, which is described in detail elsewhere [5]. In general, the objectives of the analysis were: (a) to analyze the tendencies of different health conditions, based on data from previous surveys, applying asymptotic z test; (b) to estimate the average of affiliation by different health services. All indicators were estimated considering the effect of the design and the corresponding expansion factors.

Additionally, an ordinal logistic regression model was used to evaluate distributional changes in the prevalence of overweight and obesity for males and females 20 years old and older from 2016 survey at different food insecurity levels. The model included age, social program inclusion (PROSPERA), health security affiliations, and socioeconomic index. All analyses were performed using STATA 15, and the module SVY was used to consider the complex sampling structure and the expansion factors.

Finally, a table is presented summarizing some public policies implemented by the country and referred to ENSANUT, when facing the magnitude of overweight and obesity.

### 2.3. Ethical Aspects

All procedures were performed according to the General Health Law for Health Research. The ENSANUT protocols were submitted for approval to the Ethics, Biosecurity and Research Committees of the National Institute of Public Health. The ENSANUTs included an informed consent letter addressed to the subject of study, which sets forth the objectives of the survey and the type of information that will be collected. The consent letter contains the guidelines established by the INSP Ethics Committee. Subjects over seven and under 18 years of age were asked to give their consent before the anthropometric and sampling procedures were performed.

## 3. Results

In Mexico, a high prevalence of overweight and obesity is present in the population older than five years of age. Figure 1 shows this trend from 1988 to 2016, by urban or rural location.

Table 1, Table 2, Table 3 and Table 4 show the 2006, 2012, and 2016 estimates for the percentages of the population’s health service affiliation by gender and age group. In all the groups there is an important proportion of the population with OW + O and without affiliation to health services. Also, the results are stratified by socioeconomic tertile and by overweight and obesity.

The results show that the prevalence of overweight and obesity increased in adolescent women and adults from 2006 to 2016 (*p* < 0.01). In adolescent men and school-age girls, the prevalence stabilized above 30%, and in school-age boys we see a significant reduction in prevalence (*p* < 0.01), but it still remained above 30%.

In preschool children, the prevalence showed a decrease, but OW + O still affects around 1,000,000 children. From 2006 to 2012, health service affiliation in preschool children increased along with the OW + O prevalence. In 2016, a decrease in such prevalence was observed. By socioeconomic tertile, the prevalence of OW + O is higher in T2 and T3 compared to T1 (Table 1).

In school-age children, the prevalence of OW + O is higher in girls than in boys and has increased between surveys, independently of their affiliation. In 2006 and 2012 the prevalence increased when the tertile was higher. In 2016 this happened only in the prevalence of obesity (Table 2).

In 2016 the highest prevalence was found in adolescent women, but it also was high in men regardless of their affiliation type. By socioeconomic tertile, the prevalence increased along with the tertile (*p* < 0.01), with the exception of adolescent women for T2 vs. T3 in 2006 (*p* = 0.39), for T1 vs. T2 in 2016 (*p* = 0.42), and in men for T2 vs. T3 in 2016 (*p* = 0.09) (Table 3).

In adults the prevalence of obesity increased by year of survey (*p* < 0.05), especially in women; in spite of their affiliation, such prevalence was also high in men. The distribution in the socioeconomic tertiles was homogenous (Table 4).

A joint significant interaction between sex and food insecurity levels was found (*p* = 0.0002), where an important increase in female obesity prevalence was found at higher insecurity levels compared with males (Figure 2).

Finally, Table 5 shows examples where ENSANUT surveys were used to generate and evaluate Mexican public policies designed to reduce the social burden derived from diabetes, overweight, or obesity. A detailed presentation or evaluation of policies (Table 5) would go beyond the scope of this paper. Regarding international evaluations, the results of ENSANUT surveys are also the basis for building the indicators that monitor compliance with the Millennium Development Goals (MDGs) and the current Sustainable Development Goals (SDGs); additionally, they are also used to monitor compliance with the commitments established by the “WHO Global Strategy on Diet, Physical Activity and Health” approved by the 57th World Health Assembly.

## 4. Discussion

Mexico faces an obesity epidemic. The country has the second-highest obesity rate in the world (after the USA). Rates for adults defined as “overweight” in Mexico are 76% for women and 69% for men [19]. Furthermore, our study shows a significant association between obesity in adult women and households that suffer severe food insecurity when compared with those women without food insecurity. This could be, in part, explained by the poor diet quality in such households [29,30].

Excess body weight and food insecurity coexist [31]; household food insecurity is usually associated with excess body weight in women, but not in men [18,32].

The association between overweight and obesity and many sociodemographic variables like socioeconomic level has been previously observed in school age and teenage populations in Mexico [33].

The prevalence of overweight and obesity is one of the world’s highest, as can be seen over the years. The Ministry of Health established the National Agreement for Healthy Nutrition (ANSA), a strategy that has led to initiatives like banning sodas and regulating unhealthy food in schools, as well as the design of other initiatives that are yet to be implemented, such as a front-of-package labeling system [34].

Other strategies include taxation of sugar-sweetened beverages, improvement of standards for healthy foods in schools, and regulation of food and beverage marketing to children [34].

The role of the health system in the prevention of overweight and obesity is fundamental. A consensus exists in our country that the health system plays a fundamental role in overweight and obesity prevention [35].

Likewise, we consider it fundamental to return to the traditional Mexican cuisine that contains high-quality protein from the corn and bean combination—a meal that provides essential nutrients without additional ingredients such as fats and sugar [35], which are responsible for overweight and obesity in Mexico.

In Mexico, the main contribution of the ENSANUTs is to support the planning of the government’s National Health Programs; the evidence they provide has been fundamental to the development of public policy by different government bodies.

For several decades, the ENSANUTs in Mexico have generated consistent and relevant information for public health policy-making, and have fostered actions linked to priority themes in the public agenda. The ENSANUTs have become a required reference material to understand the country’s health challenges (with respect to conditions and access) in order to design public policy responses and anticipate future challenges based on this observation.

The ENSANUTs have great strengths, recognized experience, as well as trained and standardized personnel; with the information provided by the surveys, we can establish the trends for a significant problem such as obesity, which is at epidemic proportions in our country. Through a systematized logistical and technological effort, the ENSANUTs have achieved a conceptual design that generates trustworthy indicators that are comparable over time and have broad coverage and an important number of variables to find associations with obesity. They also support public policies in matters pertaining to health and nutrition and provide evidence on health and the macro-determinants of overweight and obesity.

One aspect of the ENSANUTs that is worth highlighting is that the databases and their documents are available to the public, as a matter of transparency, at: https://ensanut.insp.mx/.

The ENSANUT has some limitations. The surveys have been financed by the Ministry of Health; however, they do not have a specific budget for this purpose and each time the survey is conducted there is a long negotiation period that generates uncertainty. Regarding the technical limitations of the ENSANUT surveys to propose policies for the control of obesity, we can mention the following: ENSANUT surveys do not ask about the effects of policies on each individual; for instance, the consequences of the increase in the tax on sugary drinks are not recorded individually. Additionally, ENSANUT surveys are a sequence of cross-sectional studies; therefore, the variability generated by the change of sample in each survey is not controlled. Finally, non-replicable variations occur in each ENSANUT survey, and their effect is difficult to quantify in series of surveys with few repetitions.

## 5. Conclusions

The National Health and Nutrition Surveys have established the population’s situation and have become a necessary reference guide to understand the health challenges in Mexico. They have fostered actions on priority topics on the public agenda, in order to design public policy responses to those challenges and to monitor health programs. The situation of overweight and obesity invites all actors—government, academics, and civil society—to join efforts to design an effective national policy for obesity prevention.

Based on this, we consider that it is necessary to create an integral, multi-sector, and multi-level policy that allows us to contain and reverse the obesity epidemic [12,35].

The first step is to set up a surveillance system that allows us to observe the changes in overweight and obesity prevalence over time. This has been accomplished through the National Health and Nutrition Surveys in Mexico (ENSANUTs) series.

Some suggestions derived from these results: action is required, from citizens as well as from the government, to increase physical activity in a secure manner in public areas as well as work and school spaces; and to have free pure water, and access to fruits, vegetables, and whole grains. In addition, the food industry needs to be involved in producing nutritious products that do not harm health, such as legumes, whole grain cereals, and fiber.

Also, it is important for a country to have a food labeling system that is useful and easily understood, to promote nutritional and health education, to promote exclusive breastfeeding up to six months of age, and to favor adequate complementary feeding from this age.

The Mexican government must recognize that nutritional interventions at the individual level and the creation of physical activity guidelines have not worked up to now, and that overweight and obesity reduction will be accomplished only if a state strategy exists and is enforced by the law [35].

## Figures and Tables

**Figure 1 nutrients-11-01727-f001:**
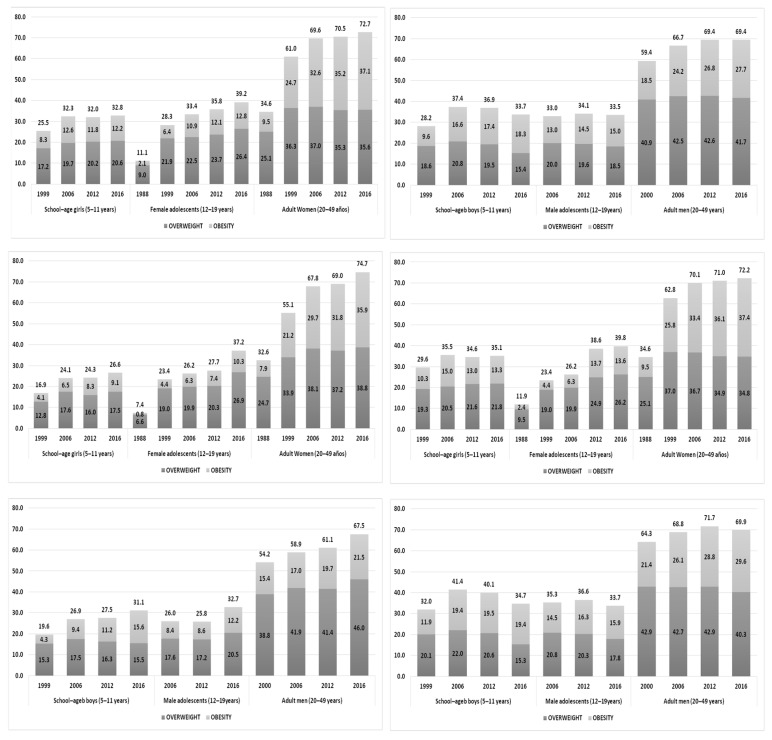
Overweight and obesity prevalence in school-age children, adolescents, and adults from 1988 to 2016: distribution by sex and locality. ENSANUT, Mexico. Anthropometric data analyzed according to WHO Growth reference data for 5–19 years (2007) Source: ENN 88 and 99, ENSANUT 2006 and 2012, ENSANUT Medio Camino, 2016.

**Figure 2 nutrients-11-01727-f002:**
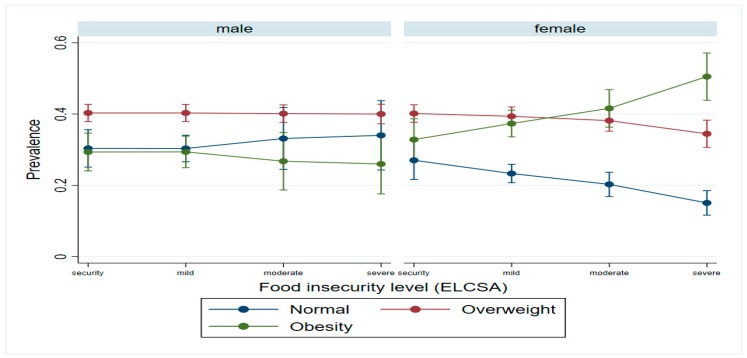
Logistic regression models * to measure the association between overweight and obesity and food Insecurity in Mexico, ENSANUT MC 2016. * Models adjusted by: age, sex, health services affiliation, socioeconomic level, and whether or not one is a beneficiary of the social development PROSPERA program.

**Table 1 nutrients-11-01727-t001:** Prevalence of overweight and obesity in the preschool population, by health service affiliation and socioeconomic tertile. 2006, 2012, and 2016 ENSANUT.

Affiliation to Health Services	2006			2012			2016		
	Total	Overweight & Obesity		Total	Overweight & Obesity		Total	Overweight & Obesity	
	Expansion			Expansion			Expansion		
	*n* (Thousands)	%	95%CI	*n* (Thousands)	%	95%CI	*n* (Thousands)	%	95%CI
None	5482.3	7.9	(6.8,9.1)	2850.2	10.4	(8.7,12.4)	1198.4	3	(0.9,9.3)
Mexican Institute of Social Security (IMSS)	2174.1	9.1	(7.3,11.3)	2760.6	9.6	(8.1,11.3)	3100.5	6.4	(3.7,10.8)
Institute of Social	268.9	9.3	(5.5,15.3)	288.2	10.3	(6.8,15.4)	682.3	4.7	(1.2,17.0)
Security for State Workers (ISSSTE)									
SEGURO POPULAR *	1165.4	8.4	(6.5,10.8)	4733.2	9.3	(8.1,10.8)	5628.2	5.8	(4.0,8.3)
Mexican Petroleum Health service (PEMEX)	11	4.1	(0.5,25.2)	24.8	27.6	(9.2,59.0)	11.4	0	-
PRIVATE	79.7	10.8	(4.2,24.8)	36.3	2.2	(0.3,15.3)	62.9	0	-
Other	213.5	8.9	(4.0,18.5)	74.1	7	(2.2,20.0)	191.2	18.1	(2.3,67.8)
Total	9394.8	8.3	(7.5,9.3)	10,767.3	9.7	(8.9,10.7)	10,874.8	5.8	(4.3,7.7)
**Socioeconomic Tertile**									
T1	3830.2	7.5	(6.3,8.9)	3601.2	8.7	(7.4,10.3)	2839.4	5.2	(3.4,7.9)
T2	3186.6	9.1	(7.7,10.8)	3717.1	10.4	(9.0,11.9)	3824	5.4	(3.5,8.2)
T3	2355.1	8.5	(6.9,10.5)	3466.9	10	(8.4,11.9)	4258.9	6.5	(3.7,11.0)
Total	9371.9	8.3	(7.5,9.3)	10,785.1	9.7	(8.9,10.6)	10,922.2	5.8	(4.3,7.6)

* Covers individuals without social security [15].

**Table 2 nutrients-11-01727-t002:** Prevalence of overweight and obesity in the school-age population, by health service affiliation and socioeconomic tertile. 2006, 2012, and 2016 ENSANUT.

Affiliation to Health Services *	2006					2012					2016				
	Total	Overweight		Obesity		Total	Overweight		Obesity		Total	Overweight		Obesity	
	Expansion					Expansion					Expansion				
	*n* (Thousands)	%	95%CI	%	95%CI	*n* (Thousands)	%	95%CI	%	95%CI	*n* (Thousands)	%	95%CI	%	95%CI
Men															
None	4307.3	19.6	(17.4,22.0)	15.2	(13.5,17.2)	1741.1	21.2	(17.9,24.8)	11.8	(9.5,14.5)	739	12.8	(6.0,25.0)	17.3	(6.5,38.5)
Mexican Institute of Social Security (IMSS)	1886.3	22.5	(19.5,25.8)	19.9	(17.0,23.1)	2157.5	21.7	(18.9,24.8)	13.8	(11.7,16.1)	1558.2	22.8	(14.9,33.2)	10.6	(5.6,19.1)
Institute of Social Security for State Workers (ISSSTE)	284.5	26.0	(19.9,33.2)	30.0	(22.7,38.5)	366.7	24.9	(18.3,32.9)	19.3	(13.7,26.6)	547.5	42.0	(12.0,79.5)	2.1	(0.6,7.5)
SEGURO POPULAR *	1118.3	20.5	(17.4,23.9)	12.6	(10.1,15.6)	3710.2	18.2	(16.5,20.1)	10.0	(8.8,11.3)	2783.5	17.3	(13.6,21.7)	10.8	(7.9,14.7)
Mexican Petroleum Health Service (PEMEX)	21.6	29.0	(10.9,57.7)	33.8	(15.2,59.3)	15.7	12.7	(2.7,43.8)	0	-	10.6	38.2	(3.7,90.9)	61.8	(9.1,96.3)
PRIVATE	30.5	17.1	(7.2,35.2)	22.7	(8.6,47.9)	17.2	12.6	(3.3,38.0)	17.5	(4.4,49.5)	44	0	-	12.9	(1.5,59.5)
Other	184.4	22.1	(12.8,35.3)	14.2	(8.4,23.0)	105.5	28.5	(15.8,45.8)	10.8	(4.6,23.0)	13.9	0	-	0	-
Total	7832.8	20.8	(19.1,22.5)	16.6	(15.2,18.1)	8114.1	20.2	(18.8,21.6)	11.8	(10.8,12.8)	5696.8	20.5	(15.0,27.2)	10.9	(7.9,14.7)
Women															
None	4338.2	19.5	(17.6,21.6)	13.2	(11.3,15.3)	1772.3	18.6	(15.7,22.0)	19.1	(15.7,23.0)	453.8	11	(5.6,20.3)	25.6	(13.2,43.7)
Mexican Institute of Social Security (IMSS)	1864.9	21.3	(18.6,24.3)	13.8	(11.3,16.7)	2252.4	21.5	(18.9,24.3)	20.8	(18.4,23.5)	1773.3	14.2	(9.0,21.9)	24.0	(12.1,42.0)
Institute of Social Security for State Workers (ISSSTE)	316.4	16.9	(10.8,25.3)	14.0	(9.4,20.2)	325.1	21.3	(16.3,27.4)	21.9	(16.6,28.3)	431.2	16.1	(5.0,41.1)	34.9	(17.9,56.8)
SEGURO POPULAR *	1142	19.1	(15.3,23.5)	8.1	(6.0,10.7)	3834.9	18.9	(17.0,21.0)	14.5	(12.8,16.5)	3023.9	16.1	(12.6,20.4)	13.9	(11.0,17.3)
Mexican Petroleum Health Service (PEMEX)	21.7	22.5	(8.9,46.6)	32.0	(10.3,65.9)	29.5	2.3	(0.3,16.6)	8.5	(2.1,29.1)	6.1	100	-	0	-
PRIVATE	53.9	21.8	(5.8,56.0)	28.7	(12.0,54.2)	20.9	7.9	(1.9,27.5)	20.9	(4.7,58.3)	1.9	0	-	22.5	(22.5,22.5)
Other	172.4	15.0	(9.0,24.0)	6.5	(2.8,14.2)	89	12.1	(5.1,26.2)	4.7	(2.0,10.5)	127.7	10.3	(1.8,42.1)	25.3	(4.3,71.8)
Total	7909.5	19.7	(18.3,21.2)	12.6	(11.3,14.2)	8324.2	19.5	(18.1,21.0)	17.4	(16.0,18.8)	5817.9	15.1	(12.1,18.7)	19.7	(14.7,25.8)
**Socioeconomic Tertile**															
Men															
T1	3007.6	18.4	(16.2,21.0)	9.6	(8.1,11.3)	2622.3	16.1	(14.1,18.3)	8.5	(7.1,10.0)	1529.4	17.0	(11.6,24.1)	5.1	(3.1,8.4)
T2	2742.2	21.4	(19.0,24.1)	19.0	(16.7,21.5)	2742.2	21.6	(19.1,24.3)	11.1	(9.7,12.8)	1795.1	16.4	(12.4,21.2)	10.3	(7.2,14.7)
T3	2062.8	23.2	(19.3,27.6)	23.5	(20.5,26.8)	2752.2	22.7	(20.0,25.7)	15.6	(13.6,17.9)	2376	25.8	(15.0,40.6)	14.9	(8.8,24.2)
Total	7812.5	20.7	(19.1,22.5)	16.6	(15.1,18.1)	8116.7	20.2	(18.8,21.6)	11.8	(10.8,12.8)	5700.5	20.4	(15.0,27.2)	10.8	(7.9,14.7)
Women															
T1	3317.8	16.8	(15.0,18.9)	8.2	(6.7,10.1)	2611	16.7	(14.9,18.7)	10.6	(8.8,12.8)	1340.5	18.2	(13.3,24.3)	8.8	(5.7,13.4)
T2	2462.9	21.1	(18.6,23.8)	14.7	(12.9,16.7)	2760.4	19.2	(16.9,21.8)	18	(15.9,20.2)	1851.6	14.2	(10.1,19.6)	13.3	(9.3,18.7)
T3	2108.2	22.4	(19.1,26.1)	17.2	(13.8,21.2)	2956.1	22.2	(19.3,25.3)	22.8	(20.0,25.7)	2637.9	14.0	(9.4,20.4)	29.6	(20.2,41.2)
Total	7888.8	19.7	(18.2,21.2)	12.7	(11.3,14.2)	8327.4	19.5	(18.1,21.0)	17.4	(16.0,18.8)	5830.1	15.1	(12.1,18.6)	19.6	(14.7,25.8)

* Covers individuals without social security [15].

**Table 3 nutrients-11-01727-t003:** Prevalence of overweight and obesity in the adolescent population, by health service affiliation and socioeconomic tertile. ENSANUT 2006, 2012, and 2016.

Affiliation to Health Services	2006					2012					2016				
	Total	Overweight		Obesity		Total	Overweight		Obesity		Total	Overweight		Obesity	
	Expansion					Expansion					Expansion				
	*n* (Thousands)	%	95%CI	%	95%CI	*n* (Thousands)	%	95%CI	%	95%CI	*n* (Thousands)	%	95%CI	%	95%CI
Men															
None	5063.7	20.3	(18.4,22.4)	11.7	(10.1,13.6)	2506.5	17.3	(14.7,20.2)	14.1	(11.9,16.8)	1037.6	21.7	(13.9,32.2)	8.7	(4.2,17.1)
Mexican Institute of Social Security (IMSS)	2083	21.8	(19.0,24.9)	14.0	(11.9,16.5)	2402.7	22.4	(19.3,25.7)	19.5	(16.6,22.8)	3213.5	18.5	(12.0,27.4)	15.1	(10.7,21.0)
Institute of Social Security for State Workers (ISSSTE)	440.4	19.2	(12.3,28.8)	16.9	(12.0,23.2)	425.3	21.2	(16.5,26.7)	17.5	(12.7,23.5)	1082.4	7.1	(3.3,14.7)	20.7	(9.8,38.5)
SEGURO POPULAR *	1183.1	15.6	(12.7,19.0)	11.0	(7.8,15.3)	3761.5	18.8	(17.0,20.8)	11.2	(9.5,13.3)	5981.3	19.3	(15.3,23.9)	15.1	(11.3,19.9)
Mexican Petroleum Health Service (PEMEX)	22.4	60.6	(35.9,80.9)	23.2	(9.5,46.5)	32.2	42.8	(16.0,74.6)	8.6	(2.1,29.7)	77.5	0	-	15.1	(1.6,66.2)
PRIVATE	114.7	3.2	(0.7,13.2)	64.3	(23.4,91.4)	20.4	0	-	14.7	(4.3,39.8)	25.8	60.3	(12.9,93.9)	18.5	(1.9,73.1)
Other	236.9	27.2	(15.9,42.4)	6.1	(3.5,10.6)	76.4	37.0	(18.7,60.1)	15.9	(6.3,34.6)	99.4	21.2	(5.9,53.3)	0	-
Total	9144.1	20.0	(18.6,21.6)	13.0	(11.3,14.8)	9225	19.6	(18.2,21.1)	14.5	(13.3,15.8)	11,517.4	18.1	(14.9,21.8)	14.9	(12.2,18.1)
Women															
None	5267.3	21.6	(19.6,23.6)	10.7	(9.1,12.6)	2228.3	23.5	(20.3,27.1)	12.2	(10.1,14.7)	1225.3	35.8	(24.6,48.8)	11.8	(5.4,23.9)
Mexican Institute of Social Security (IMSS)	2025.1	24.2	(21.2,27.6)	12.2	(10.2,14.5)	2178.6	24.7	(21.6,28.1)	13.1	(10.9,15.6)	3570.8	24.9	(16.6,35.5)	15.5	(8.2,27.3)
Institute of Social Security for State Workers (ISSSTE)	395.1	18.6	(13.1,25.7)	14.0	(8.0,23.3)	367.6	25.5	(19.1,33.0)	16.5	(11.5,23.1)	723.9	25.2	(13.1,42.9)	21.8	(10.0,41.1)
SEGURO POPULAR *	1182.6	24.5	(21.5,27.8)	8.4	(6.4,11.0)	3989.5	23.0	(20.7,25.3)	11.1	(9.4,13.2)	5580.2	24.2	(20.7,28.2)	11.2	(8.9,14.1)
Mexican Petroleum Health Service (PEMEX)	32.8	42.9	(21.2,67.7)	13.9	(4.6,35.4)	17.1	24.4	(7.6,56.0)	5.8	(1.3,22.7)	26.1	0	-	0	-
PRIVATE	61.6	35.1	(15.7,61.1)	3.7	(1.1,11.2)	27.5	23.2	(9.4,47.0)	24	(5.7,62.5)	64.4	13.3	(1.5,60.1)	17.3	(2.0,67.6)
Other	178.5	19.8	(12.7,29.5)	12.9	(6.0,25.6)	58.9	34.4	(19.5,53.3)	7.2	(2.8,17.1)	130.1	19.2	(7.3,41.8)	12.5	(2.8,41.0)
Total	9143	22.5	(21.1,24.1)	10.9	(9.7,12.2)	8867.5	23.7	(22.1,25.5)	12.1	(10.9,13.4)	11,320.8	25.6	(21.9,29.6)	13.3	(10.2,17.3)
**Socioeconomic Tertile**															
Men															
T1	3082.8	17.6	(15.6,19.9)	7.5	(5.9,9.4)	2632	16.5	(14.3,19.0)	8.5	(6.8,10.5)	2710.3	15.1	(11.1,20.2)	9.7	(5.8,15.9)
T2	2988.8	20.7	(18.0,23.7)	13.3	(11.2,15.6)	3062.1	20.4	(18.1,22.8)	13.9	(11.9,16.2)	3626.9	17.3	(13.0,22.6)	15.9	(12.2,20.5)
T3	3061.6	21.9	(19.0,25.0)	18.3	(14.4,23.0)	3538	21.2	(18.7,24.0)	19.5	(17.1,22.1)	5215.9	20.1	(15.0,26.4)	17.0	(12.6,22.4)
Total	9133.1	20.1	(18.6,21.6)	13.0	(11.4,14.8)	9232.1	19.6	(18.2,21.1)	14.5	(13.3,15.8)	11,553.2	18.0	(14.9,21.7)	14.9	(12.2,18.1)
Women															
T1	3241.6	21.3	(19.1,23.6)	8.8	(7.2,10.8)	2575.7	22.2	(19.4,25.4)	8.6	(7.2,10.2)	2563.4	23.5	(18.9,28.8)	10.4	(7.0,15.0)
T2	3124.8	23.2	(20.7,25.9)	11.9	(9.8,14.4)	2906.3	24.0	(21.3,27.1)	13.7	(11.6,16.1)	3775.2	24.7	(20.3,29.8)	9.7	(6.4,14.4)
T3	2745.5	23.4	(20.5,26.5)	12.1	(10.0,14.6)	3388.7	24.5	(21.9,27.4)	13.4	(11.5,15.5)	5028.8	27.3	(21.0,34.7)	17.4	(11.5,25.5)
Total	9111.9	22.6	(21.1,24.1)	10.9	(9.7,12.2)	8870.7	23.7	(22.1,25.5)	12.1	(10.9,13.4)	11,367.4	25.6	(22.0,29.6)	13.3	(10.1,17.2)

* Covers individuals without social security [15].

**Table 4 nutrients-11-01727-t004:** Prevalence of overweight and obesity in the adult population, by health service affiliation and socio-economic tertile. 2006, 2012, and 2016 ENSANUT.

Affiliation to Health Services	2006					2012					2016				
	Total	Overweight		Obesity		Total	Overweight		Obesity		Total	Overweight		Obesity	
	Expansion					Expansion					Expansion				
	*n* (thousands)	%	95%CI	%	95%CI	*n* (thousands)	%	95%CI	%	95%CI	*n* (thousands)	%	95%CI	%	95%CI
Men															
None	12,157	41.5	(39.5,43.4)	22.2	(20.4,24.0)	8803.5	40.2	(37.8,42.6)	24.7	(22.7,26.9)	4395.6	41.9	(31.9,52.7)	24	(17.2,32.3)
Mexican Institute of Social Security (IMSS)	7945	44.2	(41.8,46.7)	26.5	(24.3,28.8)	10,863.9	43.2	(40.7,45.7)	30.2	(28.0,32.5)	11,241.2	41.9	(34.7,49.5)	31.9	(25.3,39.3)
Institute of Social Security for State Workers (ISSSTE)	1361	45.3	(39.1,51.6)	33.9	(27.7,40.7)	1857.7	47.5	(43.4,51.8)	31.0	(27.3,35.0)	2836	39.2	(25.3,55.1)	41.5	(26.5,58.2)
SEGURO POPULAR *	2022.6	42.2	(38.8,45.6)	20.6	(18.0,23.5)	10,206.4	42.8	(40.9,44.8)	24.0	(22.4,25.8)	14098	42.1	(38.0,46.2)	23.4	(19.9,27.3)
Mexican Petroleum Health Service (PEMEX)	70.3	57.2	(41.0,72.0)	26.5	(13.9,44.6)	135.2	41.6	(23.3,62.6)	36.3	(23.1,52.0)	43.0	46.1	(13.9,81.9)	32.8	(13.5,60.4)
PRIVATE	333.3	21.2	(9.4,41.1)	25.6	(10.9,49.2)	131.7	56.1	(37.6,73.0)	31.6	(18.6,48.2)	173.1	26.4	(8.6,57.7)	16.1	(5.0,41.3)
Other	381.3	49.4	(40.8,58.1)	24.5	(17.9,32.7)	322.6	38.2	(25.0,53.3)	30.7	(20.7,42.9)	700.5	36.7	(15.5,64.6)	20.0	(7.3,44.1)
Total	24,270.7	42.5	(41.1,44.0)	24.2	(22.9,25.5)	32,320.9	42.5	(41.2,43.8)	26.9	(25.7,28.0)	33,487.4	41.6	(37.9,45.3)	27.8	(23.8,32.1)
Women															
None	17,037.6	37.0	(35.4,38.7)	32.6	(31.1,34.1)	7496.6	33.8	(31.3,36.3)	36.8	(34.3,39.5)	5065.9	35.7	(28.5,43.8)	38.6	(31.3,46.6)
Mexican Institute of Social Security (IMSS)	10,676.3	38.0	(35.6,40.6)	36.7	(34.3,39.1)	11,672.5	36.1	(34.1,38.2)	39.2	(37.1,41.3)	12,165.8	36.8	(32.1,41.8)	38.9	(33.9,44.2)
Institute of Social Security for State Workers (ISSSTE)	2002.9	33.6	(29.7,37.7)	41.0	(36.4,45.9)	2457.7	39.0	(35.5,42.5)	35.6	(32.4,39.0)	2675.9	29.8	(21.9,39.1)	46.5	(35.1,58.3)
SEGURO POPULAR *	3527.3	37.2	(34.7,39.8)	36.0	(33.4,38.7)	13,503.7	35.5	(34.0,37.1)	36.7	(35.2,38.3)	16,031.3	38.4	(35.2,41.8)	37.3	(34.3,40.3)
Mexican Petroleum Health Service (PEMEX)	142.6	39.4	(27.2,53.1)	51.5	(34.7,68.1)	155.1	26.9	(17.2,39.5)	39.4	(25.1,55.8)	198.3	23.0	(6.6,55.7)	68.5	(32.4,90.8)
PRIVATE	323.3	54.8	(40.6,68.4)	17.6	(10.6,27.8)	107.1	40.0	(22.1,60.9)	37.8	(24.5,53.2)	181.9	46.7	(21.0,74.3)	18.1	(6.5,41.3)
Other	876	41.6	(34.7,49.0)	28.1	(21.4,36.0)	348.4	26.5	(16.6,39.4)	43.3	(30.5,57.0)	566.1	57.6	(36.8,76.0)	27.8	(12.9,50.0)
Total	34,586	37.4	(36.1,38.8)	34.5	(33.4,35.7)	35,741.2	35.5	(34.5,36.5)	37.6	(36.5,38.6)	36,885.2	37.1	(35.0,39.4)	38.6	(36.3,41.0)
**Socioeconomic Tertile**															
Men															
T1	7019.7	41.8	(39.9,43.7)	17.1	(15.6,18.7)	8193.3	40.6	(38.6,42.6)	20.7	(19.2,22.4)	7088	39.0	(33.5,44.9)	19.4	(15.9,23.4)
T2	8116	42.8	(40.5,45.1)	26.0	(24.0,28.0)	10,297.3	43.1	(41.0,45.2)	26.4	(24.7,28.2)	9952	47.7	(41.9,53.6)	26.6	(21.9,32.0)
T3	9048.1	42.9	(40.0,45.9)	28.3	(25.7,31.1)	13,901.5	43.4	(41.3,45.5)	30.7	(28.6,32.9)	16,520.5	39.2	(32.7,46.1)	31.9	(25.4,39.2)
Total	24,183.8	42.6	(41.1,44.0)	24.3	(23.0,25.6)	32,392.1	42.6	(41.3,43.8)	26.8	(25.7,28.0)	33,560.5	41.7	(38.0,45.4)	27.7	(23.7,32.0)
Women															
T1	10,560.7	37.6	(35.7,39.5)	30.7	(29.0,32.4)	8920.7	35.3	(33.7,37.1)	34.4	(32.8,36.0)	7605.4	37.9	(33.8,42.0)	37.3	(33.4,41.4)
T2	11,623.2	38.1	(36.0,40.3)	36.7	(34.8,38.7)	11,613.7	36.8	(35.1,38.6)	38.6	(36.9,40.3)	11,315.8	36.8	(32.4,41.5)	41.1	(36.7,45.6)
T3	12,265.3	36.6	(34.3,38.9)	35.8	(33.6,38.1)	15,243.7	34.5	(32.7,36.4)	38.6	(36.8,40.4)	18,039.2	37.1	(33.6,40.8)	37.5	(34.0,41.3)
Total	34,449.3	37.4	(36.1,38.7)	34.5	(33.4,35.7)	35,778.2	35.5	(34.5,36.5)	37.5	(36.5,38.6)	36,960.5	37.2	(35.0,39.4)	38.6	(36.2,41.0)

* Covers individuals without social security [15].

**Table 5 nutrients-11-01727-t005:** Public health policies implemented in Mexico and based or evaluated on the ENSANUT surveys.

Survey(s)	Finding	Policy
2000—ENSA 2006—ENSANUT 2012—ENSANUT	High prevalence of overweight and obesity in all age groups of the Mexican population, tending to rise [12,16,17,18,19].	2010. National Agreement for Dietary Health (*ANSA,* Spanish acronym) [20] 2013. National Strategy for the Prevention of Overweight, Obesity and Diabetes (*ENPCSOyD*, Spanish acronym) [21].
2006—ENSANUT	12.7% of children under five years of age suffer from short stature and low weight; in addition, 26% of children between five and 11 years old are overweight or obese [22].	2010. Guidelines for the sale and distribution of foods and beverages in the country’s elementary schools [22].
2016—ENSANUT Mid-Stage	Evaluations of people’s understanding of Front-of-Pack Labeling System for Foods and Beverages system showed great difficulty in interpreting the data contained on the labels [23,24,25,26].	2015. “Daily Dietary Guidelines” or DDG (*GDA*-Spanish Acronym) [27].
2016—ENSANUT Mid-Stage	Evaluation of the sweetened beverages tax showed that adults who knew about the tax had a higher chance (OR = 1.30) of reporting a decrease in the consumption of sweetened beverages (*p* = 0.012) [28].	2014. Implementation of the tax on sweetened beverages [28].

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
