# Peer review of "The Mexican National Health and Nutrition Survey as a Basis for Public Policy Planning: Overweight and Obesity"

_nutrients, 2019, doi:10.3390/nu11081727_

Round 1
Reviewer 1 Report
The aim of the manuscript was to describe the tendency of overweight and obesity in Mexico and to give examples of public policies derived from the results of the Mexican National Health and Nutrition Surveys (ENSANUT). The situation of overweight and obesity in Mexico invites government, academics and civil society to join efforts to design an effective national policy for obesity prevention.
Major comment
Maybe, more suggestions and solutions could be analyse by authors even if it is not the main aim of this manuscript.
Minor Comment
Please, Table 5 needs a different layout to be more easily readable.
Author Response
Response to Reviewer 1 Comments
The aim of the manuscript was to describe the tendency of overweight and obesity in Mexico and to give examples of public policies derived from the results of the Mexican National Health and Nutrition Surveys (ENSANUT). The situation of overweight and obesity in Mexico invites government, academics and civil society to join efforts to design an effective national policy for obesity prevention.
Major comment
Maybe, more suggestions and solutions could be analyse by authors even if it is not the main aim of this manuscript.
Response 1: Other solution suggestions have been included in the recommendations section.
Minor Comment
Please, Table 5 needs a different layout to be more easily readable.
Response 2: Thanks for the observation. We simplified the presentation, the table: a) it presents only 3 columns, b) now, it has the beginning year of the policies, and c) it presents up to two findings from the surveys
Reviewer 2 Report
This manuscript presents some interesting information, however I feel that there is no clear aim to the paper. What aim is this paper trying to achieve? In addition, I have some specific comments, as follows:
Introduction, page 2, line 46-53: More detail is required here. For example, the 'high prevalences of overweight and obesity' in line 46 should be quantified. Likewise, what was the prevalence of OW+O in Mexico in 2017? What is the significance of stating that "previously it had been documented that Mexico was the second OECD country" - is this indicating that levels of OW+O seem to have gone up? What year was this 'previous' estimate made?
Introduction, page 2, lines 54-59: This could be more clearly linked to the preceding paragraph.
Methods, page 2, lines 70-71: Could the "life line scheme" be explained more clearly? What does this actually mean?
Results, Figure 1. What does this figure purport to show? Is the trend across time significant? How is a 'high prevalence' defined?
Results, page 4, lines 130-136. It is stated that the 'prevalence of overweight and obesity increases... regardless of gender, affiliation type and socioeconomic status'. Is this increase significant? Has any significance testing been conducted? Likewise for the results concerning preschool children.
Results, page 11, lines 164-166. These results for food insecurity seem to appear out of nowhere. Could more context be given?
Results, table 5. I'm not sure how this information fits in. Could it be linked more clearly with the analysis?
Discussion, page 14, lines 192-202. This information needs to be more clearly linked with the results presented in this manuscript.
Discussion, page 14, lines 209-212. What is the relevance of this information?
Author Response
Response to Reviewer 2 Comments
This manuscript presents some interesting information, however I feel that there is no clear aim to the paper. What aim is this paper trying to achieve? In addition, I have some specific comments, as follows:
Response 1: The aim of the present manuscript is –as is mentioned- to describe the changes in the overweight and obesity prevalences in Mexican population, documenting this through the National Health and Nutrition Surveys, and to show examples of public policies by using the results of ENSANUTS for the overweight and obesity control in Mexico.
Introduction, page 2, line 46-53: More detail is required here. For example, the 'high prevalences of overweight and obesity' in line 46 should be quantified. Likewise, what was the prevalence of OW+O in Mexico in 2017? What is the significance of stating that "previously it had been documented that Mexico was the second OECD country" - is this indicating that levels of OW+O seem to have gone up? What year was this 'previous' estimate made?
Response 2: We included in line 47 the quantification. It is known, that OECD estimations are based on countries that belong to it. In the case of Mexico are based on the National Health and Nutrition Surveys and when such estimations are done with the other countries are published one year after. This means those are based on ENSANUT 2012 and 2016 and are published in 2013 and 2017 respectively.
Introduction, page 2, lines 54-59: This could be more clearly linked to the preceding paragraph.
Response 3: Thank you, we have linked it.
Methods, page 2, lines 70-71: Could the "life line scheme" be explained more clearly? What does this actually mean?
Response 4: We have added information in order to clarify it.
Figure 1. What does this figure purport to show? Is the trend across time significant? How is a 'high prevalence' defined?
Response 5: Thank you. We have added information in order to clarify it.
Results, page 4, lines 130-136. It is stated that the 'prevalence of overweight and obesity increases... regardless of gender, affiliation type and socioeconomic status'. Is this increase significant? Has any significance testing been conducted? Likewise for the results concerning preschool children.
Response 6: The results show that the prevalence of overweight and obesity increases in adolescents and adults from 2006 to 2016, regardless of gender, affiliation type and socioeconomic status. In school-age children we see that the prevalence has stabilized, but reaches more than 30%. We have added information in order to clarify it.
Results, page 11, lines 164-166. These results for food insecurity seem to appear out of nowhere. Could more context be given?
Response 7: By mistake, the food security variables were not included in methodology section. They have been included. Thank you.
Results, table 5. I'm not sure how this information fits in. Could it be linked more clearly with the analysis?
Response 8: Table 5 is presented because the study’s objective is to show the overweight and obesity magnitude in Mexico and actions for public policy. We included in the analysis section a phrase stating that a brief outline of the public policies implemented in Mexico is included, this using information from ENSANUT.
Discussion, page 14, lines 192-202. This information needs to be more clearly linked with the results presented in this manuscript.
Response 9: These paragraphs were moved and linked to the study’s results because we meant to provide an example of how ENSANUT generates evidence on health and nutrition status in Mexican Population.
Discussion, page 14, lines 209-212. What is the relevance of this information?
Response 10: We have completed the idea and rewrite it with another wording. Thank you very much for the comment because the phrase was not well understood.
Round 2
Reviewer 2 Report
Thank you for your responses. I have just a few further comments.
Response 2: The years in which these estimations are done should be included in text. I.e. Line 50 should read "But previously (2013) Mexico was the second OECD country..." Also, as in my original comment, can the authors state (in text) the significance of this statement - is this indicating that levels have gone up, or that Mexico has risen in the ranks, or both? What was the prevalence in 2017?
Results, lines 170-179: Can the significance of these increases in prevalence be stated?
Author Response
Thank you for your responses. I have just a few further comments.
Response 2: The years in which these estimations are done should be included in text. I.e. Line 50 should read "But previously (2013) Mexico was the second OECD country..." Also, as in my original comment, can the authors state (in text) the significance of this statement - is this indicating that levels have gone up, or that Mexico has risen in the ranks, or both? What was the prevalence in 2017?
Response 1: Thank you. We included the corresponding year and the prevalence of obesity as you suggested. OECD does not present significant test for this estimates.
Results, lines 170-179: Can the significance of these increases in prevalence be stated?
Response 2: Thank you. We included significance tests to complete these paragraph.